# Membrane-anchored PrP^Sc is the trigger for prion synaptotoxicity

**Jean R. P. Gatdula, Robert C. C. Mercer¤, Jose Andres Alepuz Guillen, Janelle S. Vultaggio, David A. Harris***

Department of Biochemistry and Cell Biology, Boston University Chobanian & Avedisian School of Medicine, Boston, Massachusetts, United States of America

¤ Current address: Department of Biochemistry & Molecular Genetics, College of Graduate Studies, Midwestern University, Glendale, Arizona, United States of America

* daharris@bu.edu

## Abstract

The mechanism by which prions composed of PrP^Sc cause the neuropathological aberrations characteristic of prion diseases remains elusive. Previous studies have defined a synaptotoxic signaling pathway in which extracellular PrP^Sc stimulates NMDA receptor-mediated Ca^{2+} influx, activation of p38 MAPK, and collapse of the actin cytoskeleton in dendritic spines, resulting in functional decrements in synaptic transmission. However, these studies did not determine whether synaptotoxic signaling is directly linked to conversion of cell-surface PrP^C to PrP^Sc, or whether it can be initiated by extracellular PrP^Sc independently of PrP conversion. To address this question, we employed two different experimental strategies, both of which interfere with PrP^C-PrP^Sc conversion: (1) neuronal expression of PrP^C mutants that are locked in the PrP^C conformation (G126V and V208M); and (2) application of extracellular PrP^Sc from a species (mouse or hamster) that is unable to convert neuronal PrP^C of the other species. We first confirmed that both of these strategies resulted in impaired PrP^C-PrP^Sc conversion in cultured N2a and CAD5 cell lines. To assay synaptotoxicity, we then used lentiviral transduction to express the PrP^C variants in primary cultures of hippocampal neurons from PrP-null mice, and quantitated dendritic spine density after exposure to purified prions. Expression of G126V PrP completely prevented spine retraction in response to three different murine prion strains (RML, 22L, and ME7), while the effect of V208M PrP was strain-dependent, consistent with partial stabilization of PrP structure by this mutation. Expression of hamster PrP^C or mouse PrP^C greatly attenuated spine retraction in response to murine 22L and hamster 263K prions, respectively. These findings support a model in which newly formed PrP^Sc at the neuronal surface is required to initiate prion-mediated synaptotoxic signaling. This work also suggests use of the G126V mutation as part of a therapeutic strategy to reduce PrP^Sc conversion in prion diseases.

**Data availability statement:** All relevant data are in the manuscript and its Supporting information files.

**Funding:** This work was supported by the National Institutes of Health grant number 5R01NS065244, awarded to DAH. Funders did not play a role in the study design, data collection and analysis, decision to publish, or preparation of the manuscript.

**Competing interests:** The authors have declared that no competing interests exist.

## Author summary

Prion diseases are fatal neurodegenerative disorders that affect both humans and animals. These diseases are caused by PrP$^{Sc}$, a misfolded and infectious isoform of the normal cellular prion protein (PrP$^C$), which propagates by a self-templating mechanism. While considerable progress has been made in understanding prion propagation, strain diversity, and infectivity, the early cellular events that initiate prion-induced neurodegeneration remain poorly defined. In our previous work, we used a specialized neuronal culture system to dissect a synaptotoxic signaling cascade triggered by PrP$^{Sc}$. Here, we focused on the initial events required to initiate this cascade on the neuronal surface, particularly the role of the PrP$^C$-PrP$^{Sc}$ conversion process. We demonstrate that impairing generation of newly formed, membrane-anchored PrP$^{Sc}$ on the neuronal surface prevents the synaptotoxic effect of prions, as assayed by quantitation of postsynaptic dendritic spines on cultured hippocampal neurons. Our results demonstrate that membrane-attached PrP$^{Sc}$ is the proximate trigger for prion-induced neurodegeneration, and they suggest a novel therapeutic approach to preventing prion toxicity using PrP mutations that lock PrP into the PrP$^C$ conformation.

## Introduction

Prion diseases, or Transmissible Spongiform Encephalopathies (TSEs), are rare neurodegenerative disorders caused by accumulation in the brain of PrP$^{Sc}$, the misfolded prion protein [1–3]. Prion propagation occurs by a templated refolding process in which PrP$^{Sc}$ seeds conversion of PrP$^C$, the normal, cellular form of the prion protein, into additional molecules of PrP$^{Sc}$ [4–7]. Over the past 30 years, the mechanisms underlying prion formation, infectivity, structure, and strain properties have been well established. However, the processes by which PrP$^{Sc}$ damages neurons and causes the neuropathological aberrations characteristic of these diseases remain elusive. Understanding prion neurotoxicity at a cellular and molecular level has been hampered, in part, by the absence of *in vitro* model systems that mimic the earliest steps of neurotoxic signaling [8–12].

In addition to serving as a precursor to PrP$^{Sc}$, PrP$^C$ is essential for the process by which prions induce neurotoxic changes in the central nervous system (CNS). The first experimental evidence that prion neurotoxicity requires expression of PrP$^C$ derived from experiments in which neural tissue grafts overexpressing PrP$^C$ were introduced into the brains of PrP-deficient mice [13]. The grafts accumulated high levels of PrP$^{Sc}$ that spread into adjacent PrP-deficient tissue, but only the grafts underwent histopathological changes associated with neurodegeneration. In addition, it was reported that ablation of endogenously expressed neuronal PrP$^C$ in prion infected mice reversed early pathological, neurophysiological and behavioral changes, ultimately preventing neuronal loss and progression of disease [14,15]. This demonstrated that endogenous PrP$^C$ expression is crucial for prion-induced

neurotoxicity. Subsequent studies revealed that PrP$^C$ must be tethered to the cell membrane via a glycosylphosphatidyli-nositol (GPI) anchor to induce typical prion pathology. Mice expressing an anchorless form of PrP$^C$ accumulated extensive amyloid deposits of PrP$^{Sc}$ after prion inoculation, but displayed minimal spongiosis [16–18]. This experiment demonstrated that the removal of the GPI anchor of PrP$^C$ fundamentally alters prion pathogenesis, even though the unanchored PrP$^C$ can still function as a substrate to bind and propagate PrP$^{Sc}$.

Taken together, these previous studies suggest that PrP$^C$ on the neuronal cell surface is essential for generating a neu-rotoxic signal in response to exogenous PrP$^{Sc}$. However, it remains unclear exactly how PrP$^C$ functions in this process. On the one hand, conversion of PrP$^C$ to membrane-anchored PrP$^{Sc}$ may be the essential event, with newly generated PrP$^{Sc}$ then serving as the trigger that initiates a neurotoxic signaling cascade. On the other hand, it is possible that PrP$^C$ serves to bind and concentrate extracellular PrP$^{Sc}$ on the cell surface, with these bound molecules of PrP$^{Sc}$ functioning as neuro-toxic ligands. Consistent with these two models, there is evidence that prion propagation is a two-step process, starting with tight but relatively non-selective binding of PrP$^{Sc}$ to PrP$^C$, followed by a conversion reaction that is strongly influenced by compatibility of the amino acid sequences of the PrP$^C$ and PrP$^{Sc}$ molecules [19–24].

To determine whether conversion of cell-surface PrP$^C$ to PrP$^{Sc}$ is essential for prion neurotoxicity, we have employed two different strategies. First, we have used PrP mutants that are locked in the PrP$^C$ conformation and are relatively resis-tant to conversion to the PrP$^{Sc}$ conformation. Second, we have taken advantage of the species barrier between mouse and hamster by testing the ability of mouse PrP$^C$ to mediate neurotoxicity caused by hamster PrP$^{Sc}$, and *vice versa*. As a readout of prion neurotoxicity, we have used an experimental system developed previously, in which PrP$^{Sc}$ causes a rapid, PrP$^C$-dependent retraction of dendritic spines on cultured hippocampal neurons [25,26]. This system replicates the earliest events of prion synaptotoxicity seen *in vivo*.

## Results

### Characterization of conversion-resistant PrP mutants in transfected cell lines

We chose two PrP mutations that impair the conversion of PrP$^C$ into PrP$^{Sc}$ (S1 Fig). One of these, G127V, is a naturally occurring polymorphism in PrP that was originally identified in Kuru-resistant individuals in Papua New Guinea (S1 Fig) [27–32]. The other, V209M, was identified based on its ability to thermodynamically stabilize PrP$^C$ [33], and was subse-quently found to be a naturally occurring polymorphism [34].

Prior to commencing experiments in neurons, we characterized the ability of the murine equivalents of these two mutants (G126V and V208M) to propagate PrP$^{Sc}$ in N2a and CAD5 cells infected with three different mouse prion strains (RML, 22L, and ME7). Constructs encoding the mutants or wild-type (WT) PrP were transfected into clones of N2a or CAD5 cells in which the endogenous PRNP gene had been knocked-out by CRISPR-Cas9 editing. These cells were then exposed to exogenous PrP$^{Sc}$ for 24 hours, and chronic infection was assessed by western blotting after passaging 5–6 times. Since subcloning of N2a cells can cause changes in prion strain susceptibility, we compared four Prnp$^{-/-}$ clones (A3, B5, C1, and C3) for their ability to generate PrP$^{Sc}$ after reconstitution with WT PrP, and chose the clone (C1) that expressed the highest level of PrP$^C$ and the highest level of PrP$^{Sc}$ after infection with RML and 22L prions (S2A-E Fig).

Prior to infection, transfected N2a and CAD5 cells showed stable overexpression and proper cell surface localization of both WT PrP and the two mutant PrPs (Fig 1A-1D). Consistent with published reports [35,36], CAD5 cells reconstituted with WT PrP$^C$ were able to propagate all three prion strains, while N2a cells reconstituted with WT PrP$^C$ could propagate RML and 22L but not ME7 prions. The G126V mutant was completely refractory to propagation of RML and 22L prions in N2a cells and to all three prion strains in CAD5 cells (Fig 1E-1F). The V208M mutant showed reduced or absent propaga-tion of RML and 22L prions in both cell types (Fig 1E-1F). Quantitation of the western blot results is shown in S3 Fig. We conclude from these data that the G126V and V208M mutations significantly impair prion propagation of all three strains in either N2a and/or CAD5 cells, with the G126V mutant being completely refractory under all conditions tested. The fact that the V208M mutant showed some degree of conversion, depending on the cell type and prion strain, presumably reflects

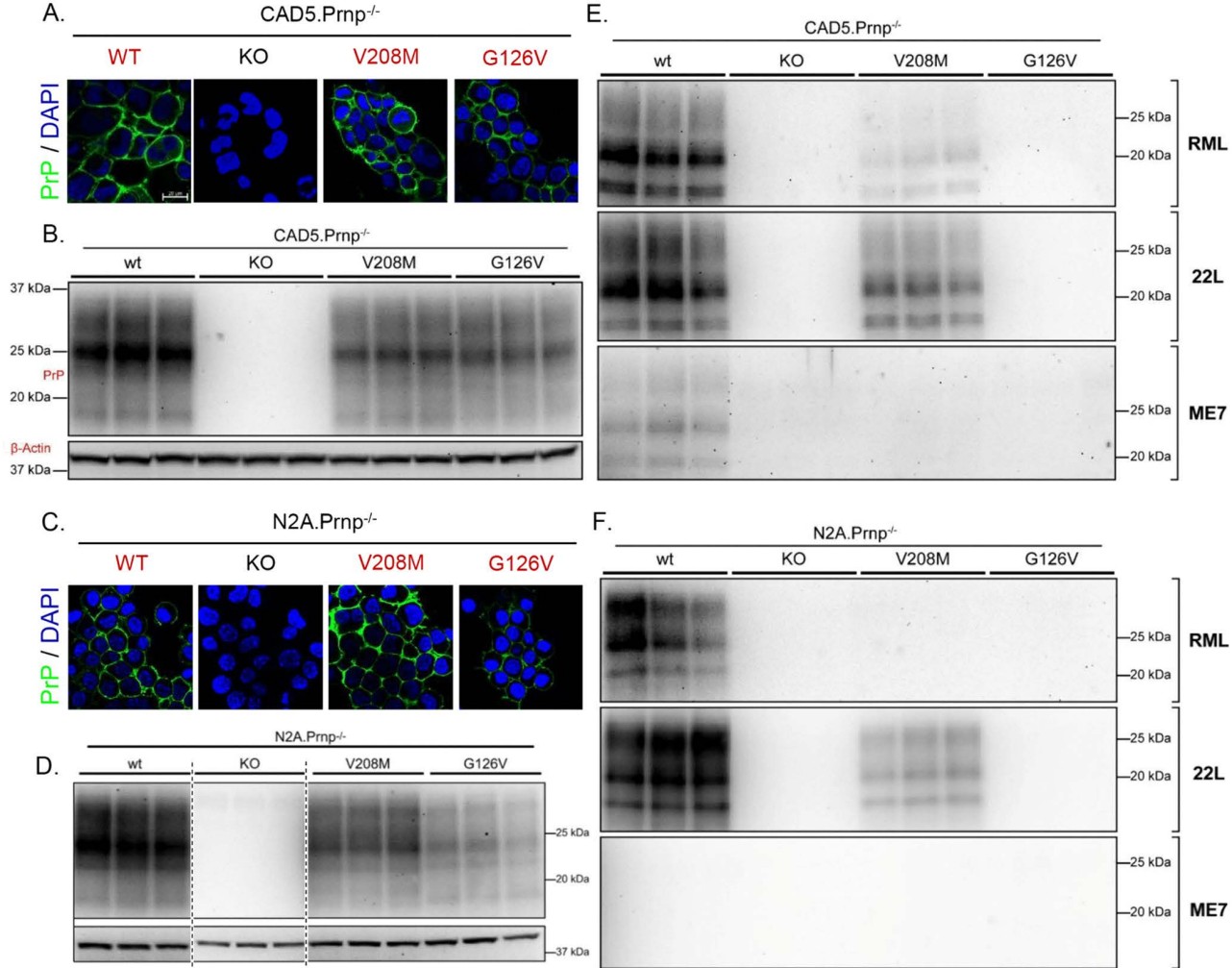

**Fig 1. Characterization of conversion-resistant PrP mutants in transfected cell lines. (A, C)** Cell surface localization of WT, G126V, and V208M PrP$^C$ expressed in N2a and CAD5 cells. Uninfected CAD5.Prnp$^{-/-}$ and N2a.Prnp$^{-/-}$ cells (C1 clone), either untransfected (KO) or expressing WT, G126V, or V208M PrP, were stained with D18 antibody (green) to detect PrP and with 4′,6-Diamidino-2-phenylindole (DAPI) (blue) to show nuclei. Scale bar = 20 μm. **(B, D)** Stable expression of WT, G126V, or V208M PrP$^C$ in the same cell lines shown in panels A and C revealed by western blotting with D18 antibody. β-actin was used as a loading control. All triplicate samples were run on the same gel and imaged at the same time. **(E, F)** CAD5 and N2a cells, either untransfected (KO) or expressing WT, G126V, or V208M PrP, were exposed to 0.1% RML, 22L, or ME7 brain homogenate for 24h and passaged 6 times to establish a chronic infection and dilute out the exogenously introduced brain inoculum. After P6, cells were lysed, proteinase-K (PK) digested, and immunoblotted for the presence of protease-resistant PrP$^{Sc}$. Triplicate lanes represent samples collected from three separate dishes of cells. Knockout N2a or CAD5 cells were used as negative controls.

partial stabilization of the PrP$^C$ conformation by this mutation compared to the G126V mutation, as well as the influence of cell- and strain-dependent factors.

## Conversion-resistant PrP mutants are defective in synaptotoxic signaling

Having characterized the conversion-resistant phenotype of G126V and V208M PrP$^C$ in transfected cell lines, we tested the ability of these mutants to mediate the synaptotoxic effects of PrP$^{Sc}$ on cultured hippocampal neurons. We made use of an experimental system developed by our laboratory in which PrP$^{Sc}$ elicits rapid (<24 hrs.) retraction of dendritic spines

and decrements in synaptic transmission, thereby recapitulating the earliest events in prion neurotoxicity seen *in vivo* [25,26]. Using this system, we have shown that prion synaptotoxicity requires cell-surface expression of PrP$^C$, and results from activation of an NMDA receptor-, p38 MAPK-dependent signaling pathway. Since our previous studies predominantly utilized a single mouse-adapted prion strain (RML), we broadened our analysis here to include 22L and ME7 strains. Each of the three strains elicited significant spine retraction in WT neurons, but not in neurons from Prnp$^{-/-}$ mice (S4A-D Fig). Spine densities on Prnp$^{-/-}$ neurons treated with PrP$^{Sc}$ from all three strains were not statistically different from those on untreated WT neurons or WT neurons treated with mock-purified material (S4D Fig).

We used lentiviral transduction to express WT, G126V or V208M PrPs in hippocampal neuron cultures from Prnp$^{-/-}$ mice. The lentiviral vector utilizes the synapsin promoter to achieve neuron-specific expression. Transduction efficiency was typically near 100%, with the encoded PrPs expressed exclusively in neurons, and not in the glial feeder layer, which was derived from Prnp$^{-/-}$ mice (S5A-B Fig). We confirmed that the lentiviral transduction procedure itself did not alter dendritic spine density (S5C-D Fig).

We tested the ability of Prnp$^{-/-}$ neurons reconstituted with WT, G126V and V208M PrPs to mediate spine retraction in response to RML, 22L, or ME7 PrP$^{Sc}$. As expected, neurons expressing WT PrP showed significant spine loss compared to neurons without PrP (Fig 2A-2C). Notably, the G126V mutation dramatically reduced spine retraction triggered by all three strains, with resulting spine densities statistically indistinguishable from (22L, ME7) or only slightly less than (RML) those seen in Prnp$^{-/-}$ neurons (Fig 2A-2C). Paralleling its effect in N2a and CAD5 cells, the V208M mutation prevented spine retraction less completely, and in a strain-dependent fashion. It afforded almost complete protection with RML, minimal but statistically significant protection with ME7, and no significant protection with 22L (Fig 2C).

## G126V PrP does not support acute formation of cell-surface PrP$^{Sc}$ or chronic propagation of PrP$^{Sc}$ in neurons

Our experiments with N2a and CAD5 cells demonstrated that PrP$^C$ carrying the G126V mutation is completely refractory to prion conversion after multiple cell passages. To demonstrate that this mutant is also unconvertible on the surface of neurons within the same time-frame as our dendritic spine retraction assay, we utilized immunofluorescent staining to visualize newly formed PrP$^{Sc}$ aggregates on the plasma membrane. We took advantage of the known resistance of cell-surface PrP$^{Sc}$ to GPI anchor cleavage by phosphatidylinositol-specific phospholipase C (PIPLC) to distinguish it from PIPLC-sensitive PrP$^C$. After PIPLC treatment, neurons were fixed and treated with GdnHCl to denature PrP$^{Sc}$ and reveal epitopes for antibody staining.

We lentivirally transduced Prnp$^{-/-}$ hippocampal neurons to express WT or G126V PrP and treated them for 24 hours with purified 22L PrP$^{Sc}$. The amount of PrP$^{Sc}$ was chosen to minimize staining of PrP$^{Sc}$ molecules derived from the inoculum, thereby allowing detection of nascent PrP$^{Sc}$ derived from endogenous PrP$^C$. As expected, neurons expressing WT PrP showed high-intensity fluorescent puncta, indicative of newly converted PrP$^{Sc}$ molecules on the cell surface (Figs 3A, 3B, and S6A). In contrast, G126V PrP-expressing neurons showed very few fluorescent puncta, indicating a failure to generate GPI-anchored PrP$^{Sc}$ (Figs 3A, 3C, and S6B). From these results, we conclude that rapid conversion of cell surface PrP$^C$ to PrP$^{Sc}$ is required for prion synaptotoxicity.

To test whether G126V PrP supports chronic propagation of PrP$^{Sc}$, we employed higher density cultures of embryonic cortical-hippocampal neurons to provide sufficient protein for biochemical analysis. Neurons from Prnp$^{-/-}$ mice were lentivirally transduced to express WT or G126V PrP and then exposed to purified 22L PrP$^{Sc}$ for five days; untransduced Prnp$^{-/-}$ neurons served as a negative control to estimate the backroon level of residual PrP$^{Sc}$ derived from the inoculum, which cannot be easily removed in the absence of cell passaging (S7A Fig). We found that WT PrP supported substantial propagation of new PrP$^{Sc}$ above the background level, whereas G126V PrP neurons retained only inoculum derived PrP$^{Sc}$ (S7B Fig). Taken together, the immunofluorescence and biochemical results demonstrate that G126V PrP is unable to sustain either acute or chronic prion infection in hippocampal neurons.

PLOS Pathogens

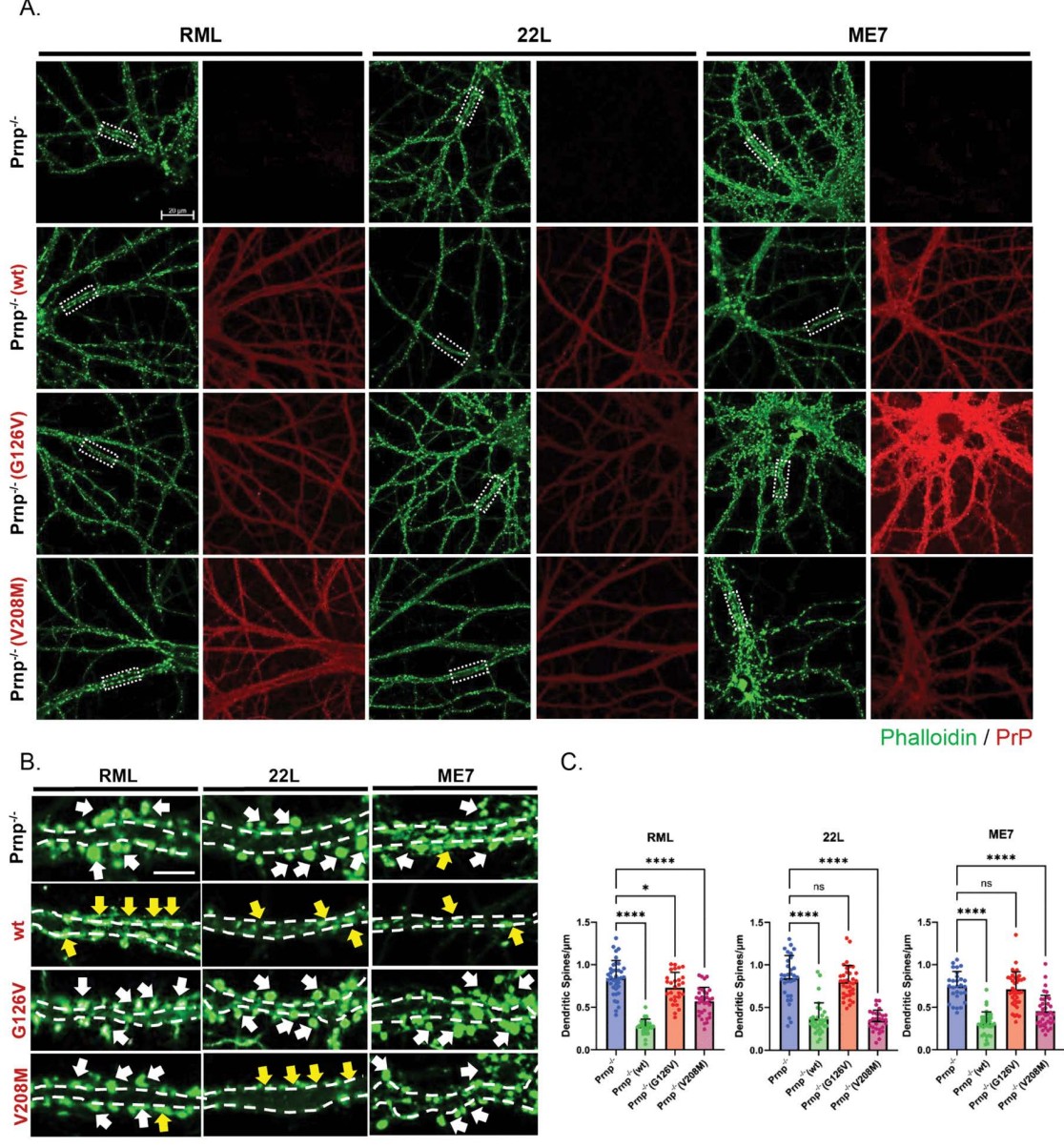

**Fig 2. Conversion-resistant PrP mutants are defective in synaptotoxic signaling.** Hippocampal neurons from ZH3 (Prnp$^{-/-}$) mice reconstituted with WT, G126V, or V208M PrP using lentiviral transduction were treated with 4.4 µg/mL of purified PrP$^{Sc}$ (RML, 22L, or ME7) for 24 hours. Cultures of non-transduced Prnp$^{-/-}$ neurons not expressing PrP were used as negative controls. **(A)** Immunofluorescent staining of WT, G126V, or V208M PrP along dendrites of transduced neurons using D18 antibody (red). Cells were also stained with Alexa488-phalloidin (green) to reveal F-actin in dendritic spines. Scale bar = 20 µm. **(B)** Higher magnification images of the boxed regions in panel A. White arrows point to dendritic spines and yellow arrows point to spines that have retracted. Scale bar = 5 µm. **(C)** Dendritic spines of 12-15 neurons (each having 3-5 dendrites/neuron) from 2 independent experiments were counted in randomly selected areas, and statistical comparisons made using a one-way ANOVA multiple comparison test. Spine number is expressed per µm length of dendrite. $p < 0.0001$ = ****; $p < 0.05$ = *; ns = not significant.

## Mouse-hamster species barrier prevents prion synaptotoxicity

As a second approach to testing whether conversion of PrP$^C$ to PrP$^{Sc}$ is necessary for transduction of a synaptotoxic signal, we took advantage of the species barrier to prion transmission, which results from the inefficiency of the PrP$^C$-PrP$^{Sc}$

PLOS Pathogens

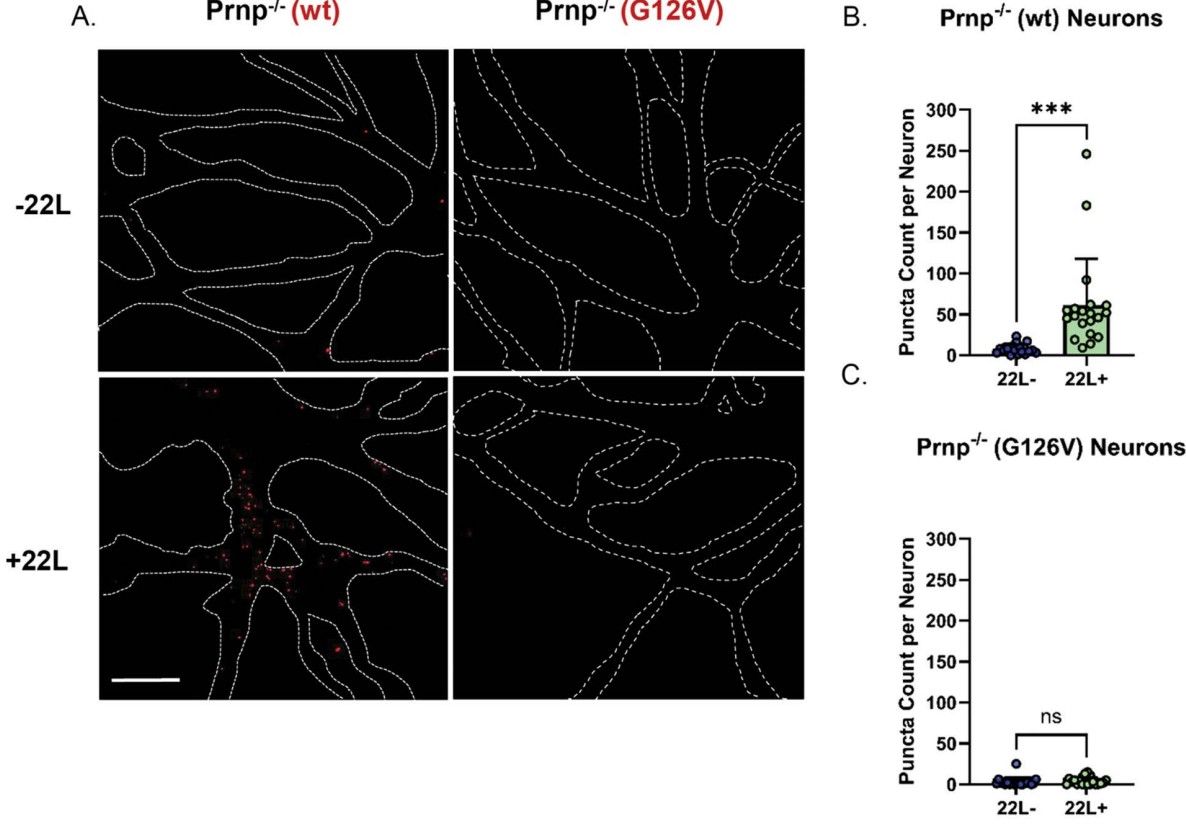

**Fig 3. Absence of prion aggregates on the surface of neurons expressing G126V PrP. (A)** Hippocampal neurons from ZH3 (Prnp$^{-/-}$) mice recon-stituted with WT or G126V PrP using lentiviral transduction were treated with 4.4 µg/mL of purified 22L PrP$^{Sc}$ for 24 hours (+*22L*). Control cultures were not treated with PrP$^{Sc}$ (-*22L*). Neurons were then stained to reveal puncta of newly converted PrP$^{Sc}$ (red) as described in Materials and Methods. These images were processed from the raw images shown in S6A and S6B Fig. Dashed white lines represent the outlines of underlying neurons, as visualized by staining with phalloidin-rhodamine (shown in S6A and S6B Fig). **(B, C)** The number of PrP$^{Sc}$ puncta was quantitated on 10 neurons each from 4 inde-pendent experiments. Puncta were visualized in randomly selected areas, and statistical comparisons made using a two-tailed student's t test. For WT neurons: -22L vs +22L, ***significantly different (p = 0.0002). For G126V neurons: -22L vs +22L, ns = not significant (p = 0.5428).

conversion process when the primary structures of the two isoforms differ. We focused on the well-characterized mouse-hamster transmission barrier, which prevents transmission of prions between the two species due to the 12 amino acid difference between mouse and hamster PrP$^C$ (S1 Fig).

We first tested the convertibility of mouse and hamster PrP$^C$ re-introduced into Prnp$^{-/-}$ N2a and CAD5 cells after expos-ing the cells to a mouse-adapted prion strain (22L) or a hamster-adapted prion strain (263K). Expression and localization of PrP$^C$ was confirmed by immunofluorescence staining and western blotting using D18 antibody, which recognizes both mouse and hamster PrP$^C$, and 3F4 antibody, which is specific for hamster PrP$^C$ (Figs 4A-4B and S8A-B). We confirmed that CAD5 cells expressing mouse PrP$^C$ could be infected with 22L but not 263K prions, while cells expressing hamster PrP$^C$ could be infected with 263K but not 22L prions (Fig 4C). Confirming previous observations [37], we found that N2a cells were unable to propagate hamster 263K prions, even after reconstitution with hamster PrP$^C$, presumably reflecting cell-type specific factors limiting formation or accumulation of hamster PrP$^{Sc}$ in this mouse cell line (S8C Fig).

We then tested whether mouse-hamster PrP$^C$ mismatch prevented dendritic spine retraction in hippocampal neurons exposed to mouse or hamster prions. We confirmed expression of mouse and hamster PrP$^C$ after lentiviral transduction using D18 and 3F4 antibodies (Fig 5A). We then challenged the transduced neurons with either mouse 22L or hamster

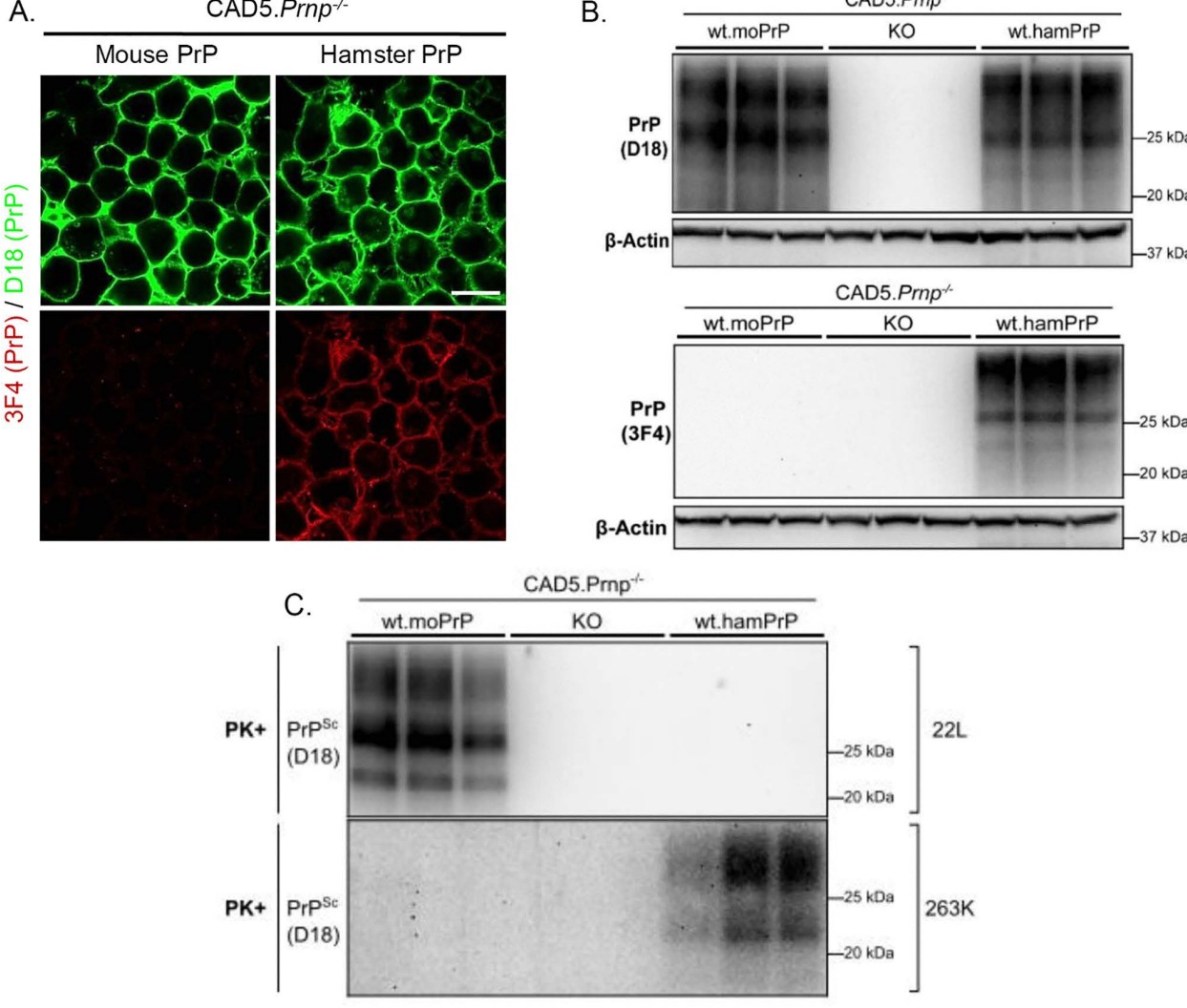

**Fig 4. Mouse-hamster species barrier prevents prion propagation in cultured cells. (A)** Cell surface localization of WT mouse and hamster PrP^C expressed in uninfected CAD5.Prnp⁻/⁻ revealed by immunofluorescent staining with D18 antibody (green), which reveals both mouse and hamster PrP^C, and with antibody 3F4 (red), which is selective for hamster PrP^C. Scale bar = 20 μm. **(B)** Stable expression of mouse (moPrP) and hamster (hamPrP) PrP^C in CAD5.Prnp⁻/⁻ (KO) cells revealed by western blotting using D18 or 3F4 antibodies. Untransfected CAD5.Prnp⁻/⁻ cells are shown as a negative control. β-actin was used as a loading control. **(C)** CAD5 cells, either untransfected (KO) or expressing mouse or hamster PrP, were exposed to 0.1% 22L or 263K brain homogenate for 24h and passaged 6 times. After P6, cells were lysed, PK digested, and immunoblotted for the presence of protease-resistant PrP^Sc. Triplicate lanes represent samples collected from 3 separate dishes of cells.

263K prions. As expected, we found that exposure of neurons to the cognate species of prion produced robust spine retraction (Fig 5B-5D). In contrast, exposure of mouse PrP^C-expressing neurons to 263K prions or hamster PrP^C-expressing neurons to 22L prions had a greatly attenuated effect on spine density (Fig 5B-5D).

Taken together with results from the previous experiments, our data demonstrate that impairing the conversion of PrP^C to PrP^Sc, whether by introduction of single amino acid substitutions that stabilize PrP^C, or by using PrP^C and PrP^Sc molecules from different species, prevents acute prion synaptotoxicity.

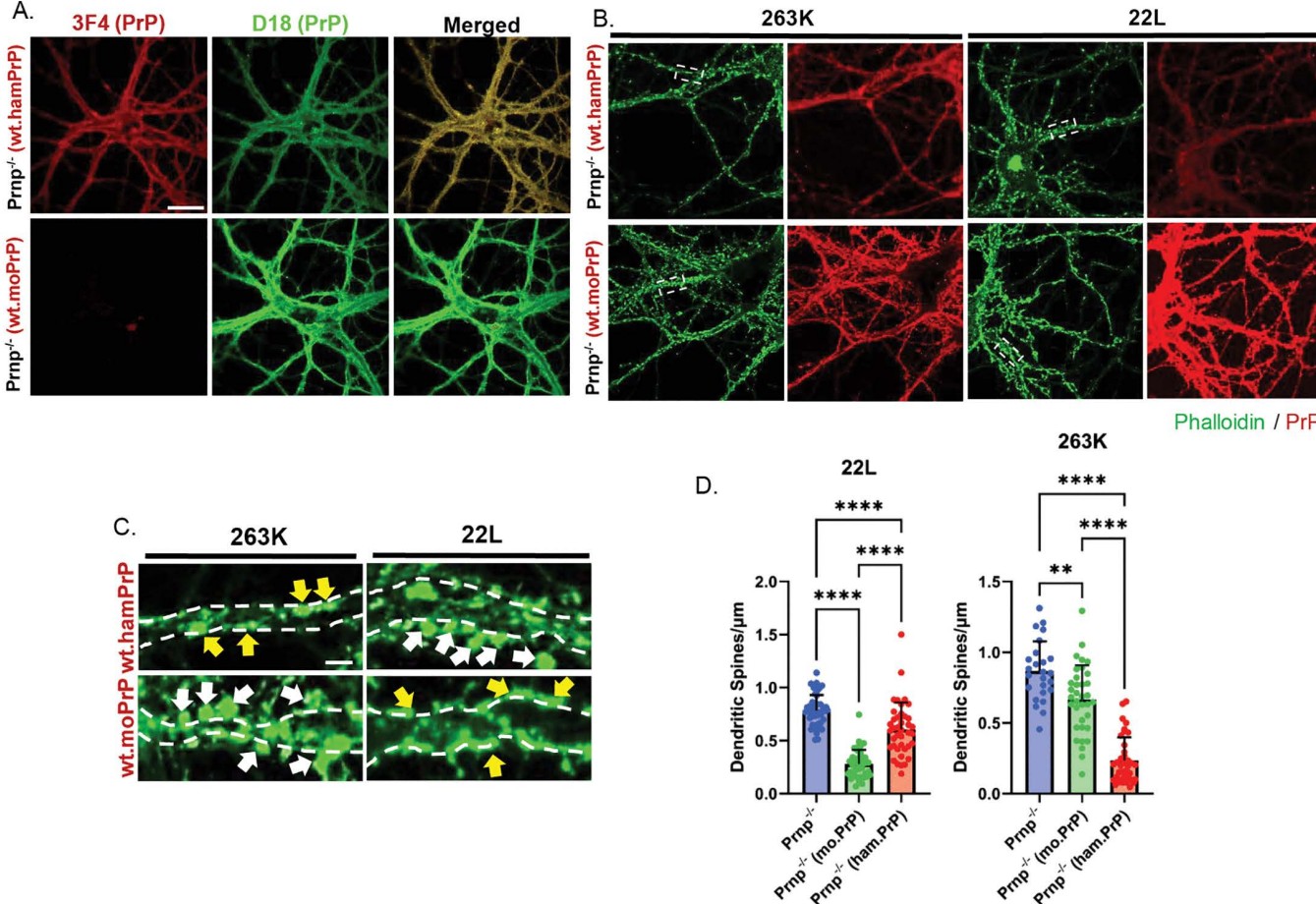

**Fig 5. Mouse-hamster species barrier prevents prion synaptotoxicity. (A)** Hippocampal neurons from ZH3 (Prnp⁻/⁻) mice reconstituted with WT mouse PrP (moPrP) or hamster PrP (hamPrP) were stained for PrP using D18 antibody (green), which detects both mouse and hamster PrP, or 3F4 antibody (red), which selectively recognizes hamster PrP. Scale bar = 20 μm (applicable to panels A and B). **(B)** Neurons were treated with 4.4 μg/mL of purified PrPSc (mouse 22L or hamster 263K) for 24 hours and then stained with D18 antibody (red) to reveal PrP, and with Alexa488-phalloidin (green) to stain F-actin in dendritic spines. **(C)** Higher magnification images of the boxed regions in panel B. White arrows point to dendritic spines and yellow arrows point to spines that have retracted. Scale bar = 2 μm. **(D)** Dendritic spines of 12-15 neurons (each having 3-5 dendrites/neuron) from 2 independent experiments were counted in randomly selected areas, and statistical comparisons made using a one-way ANOVA multiple comparison test. Cultures of non-transduced Prnp⁻/⁻ neurons treated with 22L or 263K prions were used as negative controls. Spine number is expressed per μm length of dendrite. For 22L treatments, Prnp⁻/⁻ vs mouse PrP, $p < 0.0001$; Prnp⁻/⁻ vs hamster PrP, $p < 0.0001$; mouse vs hamster PrP, $p < 0.0001$. For 263K treatments, Prnp⁻/⁻ vs mouse PrP, $p = 0.0012$; Prnp⁻/⁻ vs hamster PrP, $p < 0.0001$; mouse vs hamster PrP, $p < 0.0001$. $p < 0.0001 = ****$; $p < 0.01 = **$.

## Discussion

To test whether conversion of cell-surface PrPC into newly formed PrPSc is necessary for prion-induced synaptic degeneration, we employed a two-pronged approach: neuronal expression of PrPC molecules harboring mutations that render them non-convertible (G126V, V208M); and expression of mouse or hamster PrPC molecules in neurons that were then exposed to PrPSc from the non-cognate species. Both strategies markedly attenuated prion-induced synaptotoxicity, as measured by the extent of dendritic spine retraction on cultured hippocampal neurons exposed to purified PrPSc. Importantly, we confirmed by immunofluorescence staining that the G126V mutation prevented the appearance of PIPLC-resistant aggregates of PrPSc on the neuronal surface under the same treatment conditions used to assay spine retraction.

Taken together, these findings strongly support a model in which newly converted PrP$^{Sc}$ molecules on the neuronal surface are the proximate trigger for a synaptotoxic signaling cascade.

Our results are consistent with numerous previous studies, using genetically engineered mice, brain-grafted mice, and neuronal cultures, demonstrating that expression of PrP$^C$ in target neurons is essential for the toxic effects of prions [13,14]. Indeed, ablation of neuronal PrP$^C$ expression can actually reverse the neurodegenerative phenotype of prion-infected mice, even in the face of large amounts of astrocyte-generated PrP$^{Sc}$ [14,15]. However, none of these previous studies addressed the source of the PrP$^{Sc}$ molecules that elicit toxic changes, in particular whether they are newly formed, GPI-anchored PrP$^{Sc}$ molecules on the surface of target neurons, or, alternatively, extracellular PrP$^{Sc}$ molecules that have spread from adjacent neurons or glia. Answering this question is important for understanding the signal transduction mechanisms underlying prion pathogenesis. For example, studies of mice expressing a form of PrP$^C$ that lacks the GPI anchor display a distinctive pathology, including extensive extracellular deposits of PrP$^{Sc}$ in the brain and other tissues, often surrounding blood vessels, but lacking the spongiform changes typical of prion diseases [18]. These observations suggest that membrane-bound and extracellular PrP$^{Sc}$ may act via different pathogenic mechanisms.

A number of studies suggest that the generation of new molecules of PrP$^{Sc}$ is a two-step process, in which PrP$^{Sc}$ first binds to PrP$^C$, followed by conformational conversion of the bound PrP$^C$ to PrP$^{Sc}$. The binding step, which has been shown to involve three regions of the PrP$^C$ molecule (residues 23–33, 98–110, and 136–158 [23,38–40]), is less stringent than the subsequent conversion step. Mouse PrP$^{Sc}$ will bind to hamster PrP$^C$ and *vice versa*, even though conversion is inhibited by the mismatch in PrP amino acid sequence between the two species [39,41]. WT mouse PrP$^{Sc}$ is also likely to bind to mouse PrP$^C$ carrying the G126V and V208M mutations, since both of these substitutions lie outside of the three PrP$^C$-PrP$^{Sc}$ binding sites.

Our results imply that PrP$^{Sc}$ molecules must be generated from PrP$^C$ within the plane of the cell membrane in order to elicit a neurotoxic signal (S9 Fig). This conclusion suggests a model in which signaling results from aberrant interactions of membrane-anchored PrP$^{Sc}$ with the lipid bilayer itself or with other membrane proteins. Previous work from our laboratory has identified a prion-induced neurotoxic signaling cascade in which PrP$^{Sc}$ elicits Ca$^{2+}$ influx via NMDA receptors, followed by downstream activation of p38 MAPK, and eventually collapse of the actin cytoskeleton within dendritic spines [12,25,26]. One possible scenario is that GPI-anchored PrP$^{Sc}$ physically interacts with and activates NMDA receptors. Alternatively, we have published evidence that the polybasic N-terminal domain of PrP, which is not part of amyloid core structure of PrP$^{Sc}$ fibrils, can insert into and perturb the structure of the lipid bilayer, in some cases producing transient pores [42–44]. These changes could indirectly affect the function of NMDA receptors as well as other membrane-embedded receptors and ion channels. Models based on recent cryo-EM structures of PrP$^{Sc}$ fibrils depict how membrane-attached fibrils might distort or twist the plasma membrane [45], and this might also activate ion channels. Further work will be necessary to distinguish among these and other models.

We note that an additional test of the requirement that PrP$^{Sc}$ be GPI-anchored to elicit a synaptotoxic signal would be to express anchorless versions of PrP in neurons and test whether infection of these cells results in dendritic spine retraction. However, anchorless forms of PrP are expressed at extremely low levels in cultured cells as a consequence of ER-associated degradation [46], rendering these cells uninfectable by prions [47], and therefore making this experiment impractical.

Our results also provide interesting insights into how the structural effects of the G126V and V208M mutations influence PrP$^{Sc}$ formation in a cellular context, and the impact of strain- and cell specific factors. Prior to testing these mutants in the dendritic spine retraction assay, we characterized their behavior in cultured cells (CAD5 and N2a). In both cell types and with three different prion strains (RML, 22L, and ME7), G126V PrP$^C$ was completely refractory to conversion in this infection paradigm. This result is consistent with results obtained using CAD5 cells expressing bank vole PrP$^C$ with the G127V mutation and exposed to several mouse and hamster prion strains [48]. Interestingly, V208M PrP$^C$ was partially converted to PrP$^{Sc}$ in CAD5 cells exposed to RML and 22L, and in N2a cells exposed to 22L prions. The different behavior

of the two mutants in cell lines mirrors observations of prion-infected mice. Mice expressing exclusively G127V PrP (the human homologue of mouse G126V) are completely resistant to several CJD prion strains, and in addition, this mutant acts as a potent dominant-negative inhibitor when co-expressed with WT PrP [27]. In contrast, mice expressing V209M (the human homologue of mouse V208M) are infectible with CJD prions, but with an extended incubation time compared to WT mice [33]. NMR analyses show that the two mutations also have different structural effects on PrP$^C$ [32,33,49]. Our results demonstrate that these molecular features differentially impede PrP$^C$-PrP$^{Sc}$ conversion in cultured cells, with the G126V mutation locking the protein into a completely conversion-resistant conformation in multiple cell lines exposed to multiple prion strains.

Finally, our findings suggest use of the naturally occurring G127V polymorphism as part of a therapeutic strategy for treating prion diseases. In contrast to current efforts focused on suppressing PrP$^C$ expression via antisense oligonucleotides (ASOs) [50,51], siRNAs [52], or epigenetic editors [53], expression of G127V PrP could serve as a potent dominant-negative inhibitor of PrP$^{Sc}$ generation from wild-type PrP or PrP carrying familial prion disease mutations [27]. The kuru resistance of individuals heterozygous for this mutation [28] suggests that complete editing of the endogenous PRNP gene would not be required to achieve a therapeutic effect. It may also be possible to administer soluble, recombinant forms of G126V PrP, or engineer their expression in the brain. These approaches represent an alternative to conformational stabilization of PrP$^C$ using small molecules, ligands, or antibodies [54].

## Materials and methods

### Ethics Statement

All procedures involving animals were conducted according to the United States Department of Agriculture Animal Welfare Act and the National Institutes of Health Policy on Humane Care and Use of Laboratory Animals. Ethical approval (AN-14997) was obtained from Boston University Chobanian & Avedisian School of Medicine Institutional Animal Care and Use Committee (IACUC).

### Culture of cell lines

N2a mouse neuroblastoma cells (Product #CCL-131) were purchased from American Type Culture Collection (ATCC). N2a cells and CAD5 cells were maintained in Opti-MEM Reduced Serum Medium (Thermo Fisher #31985088) supplemented with 10% (v/v) fetal bovine serum, and 1% (v/v) penicillin/streptomycin solution (Thermo Fisher #15140122). Cells were maintained at 37 °C in a humidified incubator containing 5% $CO_2$ and were passaged every 4 days at a dilution of 1:5 using 0.25% trypsin (Fisher Scientific, MT25053 CI).

### Generation of monoclonal N2a.PrP$^{-/-}$ Cells

The ablation of endogenous PrP expressed in N2a cells was done through CRISPR-Cas9 gene editing, as described previously [55]. N2a.PrP$^{-/-}$ clones were generated using a multi-guide sgRNA system and single cell cloning by limiting dilution. 1.5 nmol of three sgRNAs (UCAGUCAUCAUGGCGAACCU, GGGCCAGCAGCCAGUAGCCA, and UCAUGGCGAACCUUGGCUAC; from Synthego) were dissolved in 15 µl of nuclease-free 1X TE buffer to make a solution of 100 pmol/µl of sgRNA. This solution was vortexed for 30 seconds and incubated at room temperature for 5 min. A final concentration of 30 pmol/µl of the sgRNAs and 20 pmol/µl of recombinant Cas9 2NLS nuclease (Synthego) were introduced to $1.0 \times 10^6$ cells by electroporation using Lonza's Amaxa Cell Line Nucleofector Kit V protocol (Program T-024). Cells were plated in 6-well dishes for recovery. After 24 hrs, or until confluence, the cells were transferred into a T-75 flask (Fisher Scientific, # FB012937) for expansion. After 3–4 days, or until confluence, the cells were trypsinized, counted, diluted to 2.0 cells/ml of medium, and plated at 500 µl/well in 48-well plates. The plates were monitored for 2 weeks for the growth of single-cell clones. Colonies were selected and expanded. N2a.PrP$^{-/-}$ clone candidates were analyzed and validated by

western blot analysis, and using Synthego's Inference of CRISPR Edits (ICE). Clones that attained a knockout score of ≥ 90% in ICE, and showed no detectable $PrP^C$ expression by western blotting, were selected and expanded further.

## Generation of stably transfected N2a and CAD5 cell lines expressing PrP constructs

The G126V and V208M mutations were introduced into wild-type mouse PrP plasmids (pcDNA3.1 (+)-Puro) through QuikChange Site-Directed Mutagenesis Kit. Primers were constructed using the QuikChange Primer Design Program: For G126V, 5'-CCCCAGCATGTAGACACCAAGGCCCCC-3' (Forward) and 5'-GGGGGCCTTGGTGTCTACATGCTGGGG-3' (Reverse). For V208M, 5'-CACATCTGCTCCACCATGCGCTCCATCATCTTC-3' (Forward) and 5'-GAAGATGATGGAGC GCATGGTGGAGCAGATGTG-3' (Reverse). Hamster PrP was introduced into the pcDNA3.1(+)-Puro vector by restriction digestion with BamHI and NheI. Wild-type hamster PrP insert containing BamHI and NheI restriction sites on the 5' and 3' ends, respectively, was generated using gBlocks from Integrated DNA Technologies (IDT).

N2a-PrP$^{-/-}$ or CAD5-PrP$^{-/-}$ cells were transfected with PrP expression constructs using Lipofectamine 3000 (Invitrogen #L3000001) according to the manufacturer's directions, or by electroporation using Lonza's Amaxa Cell Line Nucleofector Kit V protocol (Program T-024). Following transfection, cells are selected over 6–8 days in selection medium containing 2 µg/ml of puromycin. The cells are then pooled and expanded without sub-cloning. The CAD5-PrP$^{-/-}$ cells used in our experiments are monoclonal line provided by Dr. Joel Watts [37].

Prion infection of cell lines was carried out using brain homogenates from C57BL/6 Prnp$^{+/+}$ mice inoculated with RML, 22L or ME7 prions. Brain homogenates were prepared as 10% stocks in 1x PBS and used at a final concentration of 0.1% in cell culture medium.

## Hippocampal neuronal cultures

To facilitate visualization of dendritic spines, hippocampal neurons were cultured at low-density on glass coverslips suspended over a glial cell feeder layer, as described previously [25], with modifications. Briefly, hippocampi dissected from the brains of P0-P1 mouse pups (WT C57BL/6 Prnp$^{+/+}$, or Zurich 3 (ZH3) Prnp$^{0/0}$ [56]) were digested with 0.05% Trypsin in HBSS for 12 min at 37ºC. Trypsinized tissue was washed and incubated with 5 mls of plating media (DMEM/F-12, GlutaMAX supplement with 10% (v/v) FBS and 1% (v/v) penicillin/streptomycin) for 5 min at 37ºC, 5% $CO_2$. Cells were washed, resuspended, counted and seeded on poly-L-lysine coated coverslips. One hour after seeding, the coverslips were inverted onto a glial feeder layer (see below) in a 12-well plate and maintained in complete Neurobasal medium. The following day, the neuronal cultures are treated with 2µM cytosine arabinoside (AraC) to inhibit glial proliferation on the coverslips. Half-media changes were done every 3–4 days until full maturation. The neurons were kept in culture for 21–24 days *in vitro* (DIV) prior to PrP$^{Sc}$ treatment.

The glial feeder layer was generated from the brains of P1-P2 neonatal mouse pups (C57BL/6 Prnp$^{+/+}$ or Zurich 3 Prnp$^{-/-}$). Cortices were dissected in HBSS and chopped into small pieces. The chopped tissue was then incubated with digestion solution (HBSS supplemented with 1% DNAse I, 1.5 ml of 2.5% Trypsin) for 5 min at 37ºC and 5% $CO_2$. The tube was shaken 3–4 times during the incubation. The tissue was then homogenized and incubated for another 10 min at 37ºC. The homogenized tissue was then centrifuged at 1000x g for 10 min. The cell pellet was suspended in 3 ml of complete DMEM/F-12 (supplemented with 10% FBS, 1% penicillin/streptomycin, and 2% N-2 supplement) and seeded in a T-75 flask. Glia were seeded on a 12-well plate and cultured in complete Neurobasal medium (1% (v/v) GlutaMAX and 2% (v/v) B-27 Supplement) for at least two days before placement of coverslips with hippocampal neurons.

## Mixed hippocampal and cortical neuronal cultures

Hippocampi and cortices were dissected together from the brains of E16-18 Prnp$^{0/0}$ mouse embryos, followed by a 10 min digestion at 37ºC with TrypLE Express Enzyme. Tissue was washed with 5 mls of Neurobasal Plus Complete Media (Neurobasal Plus Medium, supplement with B-27 Plus Supplement, GlutaMAX and 1% (v/v) penicillin/streptomycin) and

centrifuged for 5 min at 1,000xg. Cells were washed again, resuspended in Neurobasal Plus Complete Media, counted and seeded on poly-D-lysine coated 6-well plates at a density of 250,000 cells per well. The following day, the neuronal cultures are treated with 2μM AraC to inhibit glial proliferation. Half-media changes were done twice a week. The neurons were kept in culture for 5 days in vitro prior to lentiviral transduction, followed by PrPSc treatment at DIV9. Neurons were lysed for biochemical analysis at DIV14.

### Lentiviral production and transductions

Lentiviruses encoding WT and mutant PrPs were used to transduce cultured hippocampal neurons. The EGFP coding segment in pLenti-hSYN-EGFP was exchanged for PrP coding segments to drive selective expression of PrP in neurons. To prepare viral stocks, Lenti-X HEK293T cells were plated on a gelatin-coated 15 cm diameter dishes and cultured in complete Opti-MEM (including 10% FBS and 1% penicillin-streptomycin) at 37°C in a humidified incubator containing 5% $CO_2$. After 24 h, the cells were transfected with the packaging plasmids (psPAX2 and pMD2.G) and the transfer plasmid (pLenti-hSYN-PrP) using polyethylenimine (PEI) at a 2:1 PEI:DNA ratio, and a pLenti-hSYN:psPAX2:pMD2.G ratio of 2:1.2:0.8. At 16 h post-transfection, the medium was replaced with fresh, complete Opti-MEM media. Lentivirus-containing medium was then collected 24 h and 48 h after the media change. Lentivirus containing medium was centrifuged for 900x g for 5 min, and the supernatant was then filtered through a 0.45 μm filter. For transduction experiments, lentiviral stocks were concentrated using a Lenti-X Concentrator (Takara) according to the manufacturer's directions, and then stored at -80 °C.

### Immunofluorescence staining, microscopy, and quantitation of dendritic spines

Cell lines or hippocampal neurons were washed twice with PBS for 5 min each, followed by fixation in 4% paraformaldehyde in PBS for 30 min at room temperature. Following fixation, the cells were rinsed with PBS, incubated for 5 min with 0.1M glycine in PBS, rinsed, and permeabilized with 0.1% Triton X-100 in PBS for 5 min. The cells were then incubated with blocking buffer (1% (w/v) BSA) for 30 min, and then probed with primary antibodies diluted in blocking buffer for 1hr. After rinsing with PBS, cells were incubated for 30 min with secondary antibodies conjugated to a fluorophore diluted in blocking buffer, followed by rinsing in PBS. Coverslips were then mounted with 4 μl of VECTASHIELD Antifade Mounting Medium (#H-1000–10) on SuperFrost Microscope glass slides. Images were acquired using a Zeiss AxioObserver D1 Fluorescence Microscope and/or Zeiss LSM 700 Laser Scanning Confocal Microscope with 63X oil objectives.

The following primary antibodies were used: humanized anti-PrP antibody D18 [57] (10 μg/ml), and mouse anti-PrP antibody 3F4 (1:100; Millipore Sigma, MAB1562). The following secondary antibodies were used: goat anti-mouse IgG -Alexa Fluor 647 (Invitrogen, cat. # A-21236, 1:200); goat anti-human IgG-Alexa Fluor Plus 488 (Invitrogen, cat. # A48276, 1:200). Alexa Fluor 488-conjugated phalloidin (Invitrogen; cat. A12379, 1:200) was used to stain F-actin in dendritic spines.

The number of dendritic spines was determined using the Dendritic Spine Counter plug-in in ImageJ. 3–5 dendrites were chosen for each image and the number of spines was normalized to the measured length of the dendrite to give the number of dendritic spines/μm. For each experiment, 15 neurons from 2 individual experiments from randomly selected fields of view were imaged and quantified.

### Visualization and counting of PrPSc aggregates on the surface of hippocampal neurons

After treatment for 24 h with purified PrPSc, neurons were incubated with 1.25 U/ml of PIPLC (P6466, Invitrogen) for 5 hours in complete Neurobasal media to cleave PrPC. Neurons were then washed twice with 1x PBS (with $Mg^{2+}$ and $Ca^{2+}$) and fixed with 4% paraformaldehyde in PBS for 12 min. Cells were washed once with PBS and then denatured with 3M GdnHCl for 10 min to expose antibody epitopes on PrPSc. After denaturation, cells were washed 5 times with 1x PBS and then processed for immunofluorescence staining as described above. 10–15 neurons were randomly selected and imaged using a Zeiss AxioObserver D1 Fluorescence Microscope and/or a Zeiss LSM 700 Laser Scanning Confocal Microscope with 63X oil objectives.

To facilitate counting of PrP$^{Sc}$ puncta, images were processed in ImageJ using a difference of Gaussian blurs. The PrP channel of the image was duplicated into two images. The first image was then set to have a Gaussian blur filter of 1. The second image was then set to have a Gaussian blur filter of 2. Using the Image Calculator option, the second image (Gaussian blur: 2) was subtracted from the first image (Gaussian blur: 1). The subtracted image was then adjusted to a minimum intensity of 0 and maximum intensity of 31. The threshold used to identify highly fluorescent puncta was set to a minimum intensity of 5 and a maximum intensity of 255. Image processing steps were done identically for both PrP$^{Sc}$- treated and untreated neurons.

## Live cell imaging

Live cell imaging of hippocampal neurons transduced with a synapsin promoter-driven, eGFP-encoding lentivirus (S5 Fig) was performed using an EVOS FL Auto microscope. Images were collected at 20x magnification, and analyzed with Image J processing package (Fiji).

## Immunoblotting

Cells were lysed using RIPA buffer with 0.1% SDS and adjusted with 4x Bio-Rad Laemmeli Sample Buffer (2.5% (v/v) β-mercaptoethanol). The samples are then boiled at 95ºC for 10 min and run on a 12% Criterion TGX Precast Midi Protein Gel for 45 min at 200V. Gels are transferred onto an Immobilon-P PVDF membrane for 45 min at 100V using 1X Transfer buffer. Membranes are washed with 0.1% (v/v) TBST for 5 min, and were blocked for 1 h at RT using a blocking buffer (5% (w/v) Blotto non-fat dry milk prepared in 0.1% TBST). The membrane was then probed with primary antibodies diluted in blocking buffer overnight at 4ºC. The following day, the membrane was washed with 0.1% TBST three times for 5 min each. Membranes were then probed with a horseradish peroxidase (HRP)-conjugated secondary antibody diluted in blocking buffer for 1 h at RT. The membrane was washed three times before being developed by incubating it with MilliporeSigma Immobilon Western Chemiluminescent HRP Substrate (ECL) for 2 min at RT and imaged using Bio-Rad Chemidoc XRS Molecular Imager. Quantification of band intensity was done using ImageJ. The following primary antibodies were used: human anti-PrP antibody D18 [57] (0.1 µg/ml), mouse anti-PrP antibody 3F4 (1:5000; Millipore Sigma, MAB1562), and mouse anti-actin antibody (1:10000; Millipore Sigma, A2228).

## PrP$^{Sc}$ purification

Neurons were treated with full-length PrP$^{Sc}$ purified from brain as previously described [58]. Twelve tubes of 200 µl aliquots of 10% (w/v) infected or control brain homogenate (NBH, RML, 22L, and ME7) were treated with Pronase E to give a final concentration of 100 µg/ml and incubated for 30 min at 37°C in a shaker. Samples were then treated with EDTA to give a final concentration of 10 mM. Four percent Sarkosyl in D-PBS and Benzonase were then added to give final concentrations 2% (w/v) and 50 U/ml, respectively, and incubated for 10 min at 37°C. Four percent (w/v) of NaPTA was added to each sample to give a final concentration of 0.3% (w/v), and samples were incubated for 30 min at 37°C. Samples were then mixed with iodixanol and NaPTA to give the final concentrations of 35% (w/v) and 0.3% (w/v), respectively. Samples were then centrifuged for 90 min at 16,100x g. The supernatant was then filtered using an Ultrafree-HV micro centrifuge unit (0.45 µm pore size Durapore membrane, Millipore, Prod. No. UFC30HV00) spun at 12,000x g. The filtrate was transferred to a new 1.5 ml microfuge tube and mixed with an equal volume of 2% (w/v) sarkosyl in D-PBS containing 0.3% (w/v) NaPTA pH 7.4 and incubated for 10 min at 37°C. Samples were then centrifuged for 90 min at 16,100x g. Each pellet was resuspended in 200 µl of wash buffer (17.5% (w/v) iodixanol and 0.1% (w/v) sarkosyl diluted in D-PBS), followed by the addition of NaPTA to a final concentration of 0.3% (w/v). Samples are then centrifuged at 16,100x g for 30 min at 37°C. The resulting pellet was then resuspended again in 200 µl of wash buffer followed addition of NaPTA to a final concentration of 0.3% (w/v). Samples were then centrifuged at 16,100x g for 30 min. The pellet was resuspended in 20 µl of D-PBS containing 0.1% (w/v) sarkosyl. The resuspended pellets were pooled and stored at -80ºC. Purity of the preparations was confirmed by SDS-PAGE followed by silver staining and western blotting.

## Graphics

Western blot images were labeled using Sciugo. Illustrations were made using BioRender. Graphical Abstract link: https://BioRender.com/rs71npyp; S5A Fig link: https://BioRender.com/yr3ket0. The Protein Database (PDB) numbers of the structures used to depict PrP$^C$ and PrP$^{Sc}$ in the graphical abstract are PDB: 1AG2 [59] and PDB:7QIG [60], respectively. Statistical bar graphs were made using GraphPad Prism.

## Supporting information

**S1 Fig. Sequences of WT mouse and hamster PrPs, and conversion-resistant mutants of mouse PrP.** Amino acid differences relative to WT mouse PrP are in shown in red font. Mo = mouse; Ha = Hamster.
(TIF)

**S2 Fig. Characterization of Prnp$^{-/-}$ N2a clones reconstituted with WT PrP$^C$.** Lack of PrP expression confirmed by immunofluorescent staining **(A)** and western blotting **(B)** using D18 antibody in four clones (A3, B5, C1, C3) of N2a.PrP$^{-/-}$ cells. WT PrP was introduced into the four clones of N2a.Prnp$^{-/-}$ cells and detected by immunofluorescent staining **(C)** and western blotting **(D)**. In panels A and C, DAPI is used as a nuclear counter stain (blue), and scale bar = 20 μm. In panels B and D, β-actin is used as a loading control. **(E)** Infection profile of N2a.Prnp$^{-/-}$ clones reconstituted with WT PrP and exposed to RML, 22L, and ME7 prions. Cells were treated with infected brain homogenates for 24h and then passaged 6 times. After P6, cells were lysed, PK digested, and immunoblotted for the presence of protease-resistant PrP$^{Sc}$. Duplicate lanes represent samples collected from two separate dishes of cells. Lanes labeled "IBH" represent the prion-infected brain homogenates of each strain that were used to infect the cells. The lanes labeled "N2a parent" represent the original, uncloned N2a cell population that was used to create the PrP$^{-/-}$ clones.
(TIF)

**S3 Fig. Quantitation of PrP$^{Sc}$ levels in N2a and CAD5 cells expressing WT, V208M and G126V PrP. (A, B)** ImageJ was used to quantify western blots like those shown in Fig 1E and 1F of RML, 22L, ME7 infected CAD5 and N2a cells reconstituted with WT, G126V or V208M. Samples from three independent experiments were PK-digested, immuno-blotted, and statistical comparisons made using a one-way ANOVA multiple comparison test. For all statistical analyses, PrP$^{Sc}$ bands were normalized to the PrP$^C$/actin ratios of uninfected cells. Bars show the normalized PrP$^{Sc}$ signal relative to chronically infected WT cells set to 100%. $p < 0.0001 = ****$; $p < 0.001 = ***$; $p < 0.01 = **$.
(TIF)

**S4 Fig. Three mouse adapted prion strains produce PrP$^C$-dependent dendritic spine retraction in hippocampal neurons. (A)** Hippocampal neurons from C57BL/6 Prnp$^{+/+}$ mice were treated for 24 hours with 4.4 μg/mL of purified PrP$^{Sc}$ (*RML, 22L, or ME7*). Neurons were then stained with Alexa488-phalloidin to visualize dendritic spines. Scale bar = 20 μm (applicable to main panels in A-C). The boxed regions are shown at higher magnification above the main panels. Yellow arrows point to spines that have retracted. Scale bar = 5 μm (applicable to boxed panels in A-C). **(B)** Neurons from C57BL/6 Prnp$^{+/+}$ mice were untreated or were treated for 24 hours with material that was mock-purified from uninfected brains, followed by staining with Alexa488-phalloidin. The boxed areas are show at higher magnification above the main panels. White arrows point to intact dendritic spines. There are no retracted spines. **(C)** Hippocampal neurons from ZH3 Prnp$^{-/-}$ mice were treated for 24 hours with 4.4 μg/mL of purified PrP$^{Sc}$ (*RML, 22L, or ME7*). Neurons were then stained with Alexa488-phalloidin to visualize dendritic spines. The boxed regions are shown at higher magnification above and below the main panels. White arrows point to intact dendritic spines. There are no retracted spines. **(D)** Dendritic spines of 12–15 neurons (each having 3–5 dendrites/neuron) from 2 independent experiments were counted in randomly selected areas, and statistical comparisons made using a one-way ANOVA multiple comparison test. Spine number is expressed per μm length of dendrite. $p < 0.0001 = ****$; ns = not significant.
(TIF)

**S5 Fig. Lentiviral-mediated transduction of hippocampal neurons has no effect on dendritic spine density. (A)** Hippocampal neurons from ZH3 (Prnp$^{-/-}$) mice were transduced at DIV6 with a lentivirus encoding eGFP under control of the synapsin promoter to achieve neuron-specific expression. Cultures were then imaged on DIV21 using the EVOS Live Cell Imaging Microscope with transillumination (*Trans*) or fluorescence illumination (*GFP*). *Merge* shows a superposition of the two images. The glial cells in the feeder layer (consisting of astrocytes) were plated, transduced, and imaged as a negative control to check for neuron-specific transduction of eGFP. Created in BioRender. Gatdula, J. (2026) https://BioRender.com/yr3ket0 **(B)** Prnp$^{-/-}$ neurons were untransduced, or were transduced with lentiviruses encoding WT, G126V, or V208M PrP. Cells were fixed and stained with Alexa488-phalloidin (green) to visualize F-actin in dendritic spines and with D18 antibody (red) to detect PrP. Neurons from ZH3 (Prnp$^{-/-}$) and C57BL/6 (Prnp$^{+/+}$) mice were used as negative and positive controls, respectively. Scale bar = 20 μm. **(C)** Higher magnification images of neurons showing dendritic shafts with protruding spines. Scale bar = 5 μm. **(D)** Dendritic spines of 12–15 neurons (each having 3–5 dendrites/neuron) from 2 independent experiments were counted in randomly selected areas, and statistical comparisons made using a one-way ANOVA multiple comparison test. Spine number is expressed per μm length of dendrite. Prnp$^{-/-}$ vs WT, p=0.4802; Prnp$^{-/-}$ vs G126V, p=0.9493; Prnp$^{-/-}$ vs V208M, p=0.2103. ns=not significant.
(TIF)

**S6 Fig. Images quantified in Fig 3.** Hippocampal neurons from ZH3 (Prnp$^{-/-}$) mice reconstituted with (A) WT PrP or (B) G126V PrP using lentiviral transduction were treated with 4.4 μg/mL of purified 22L PrP$^{Sc}$ for 24 hours (+22L). Control cultures were not treated with PrP$^{Sc}$ (-*22L*). Neurons were then incubated with (+*PIPLC*) or without (-*PIPLC*) 2.5 U/ml of PIPLC for 5 hours to remove PrP$^{C}$ and facilitate selective visualization of newly formed PrP$^{Sc}$ on the cell surface. After fixation, cultures were treated with 3M GdnHCl to expose antibody epitopes on PrP$^{Sc}$ prior to immunofluorescent staining using anti-PrP antibody D18 (green). Phalloidin-rhodamine was used as a counter-stain to reveal actin in neuronal processes (red). These images were processed in ImageJ using a difference of Gaussian blurs, as described in Materials and Methods, to facilitate visualization and counting of PrP$^{Sc}$ puncta (shown in Fig 3). Puncta were only detected on the surface of WT PrP-expressing neurons exposed to 22L and incubated with PIPLC prior to staining. Scale bar = 20 μm.
(TIF)

**S7 Fig. Prions cannot propagate in neurons expressing G126V PrP. (A)** Cortical-hippocampal neurons from E16-18 ZH3 (Prnp$^{-/-}$) mice were transduced on DIV 5–6 with lentiviruses encoding WT or G126V PrP; another set of cultures was not transduced (KO). From DIV 9–14, neurons were treated with 4.4 μg/mL of purified 22L PrP$^{Sc}$, after which cells were lysed and equal amounts of protein were then either left undigested (PK-) or were digested with proteinase-K digested (PK+), followed by immunoblotting to reveal PrP. Duplicate lanes represent samples collected from separate dishes of cells. Untransduced neurons (KO) were used as negative controls to estimate the PrP$^{Sc}$ signal derived from the residual PrP$^{Sc}$ inoculum that remains adhered to the cells. **(B)** Quantitation of the amount of cell-generated PrP$^{Sc}$ in the PK+ lanes after subtraction of the background due to residual inoculum in the PK+ lanes from the KO cells.
(TIF)

**S8 Fig. Hamster prions cannot propagate in Prnp$^{-/-}$ N2a cells reconstituted with hamster PrP$^{C}$. (A)** Cell surface localization of WT mouse and hamster PrP expressed in uninfected N2a.Prnp$^{-/-}$ cells (C1 clone) by immunofluorescent staining. PrP$^{C}$ was visualized with D18 antibody (green), which detects both mouse and hamster PrP, or with 3F4 antibody (red), which is specific for hamster PrP$^{C}$. Scale bar = 20 μm (applicable to all panels). **(B)** Stable expression of WT mouse (mo) and hamster (ham) PrP$^{C}$ in N2a.Prnp$^{-/-}$ cells revealed by western blotting using D18 or 3F4 antibodies. Untransfected N2a.Prnp$^{-/-}$ cells (KO) are shown as a negative control. β-actin was used as a loading control. **(C)** N2a.Prnp$^{-/-}$ cells, either untransfected (KO) or expressing mouse or hamster PrP, were exposed to 0.1% 22L or 263K prions for 24h and passaged 6 times. After P6, cells were lysed, PK-digested, and immunoblotted for the presence of protease-resistant PrP$^{Sc}$. Triplicate lanes represent samples collected from three separate dishes of cells.
(TIF)

**S9 Fig. Graphical Abstract.** Created in BioRender. Gatdula, J. (2026) https://BioRender.com/rs71npy. (TIF)

## Acknowledgments

We thank Mikel Garcia-Marcos (Boston University School of Medicine) for the gift of the Lenti-X HEK293T cells, pLenti-hSYN-EGFP transfer plasmid, and packaging plasmids (psPAX2 and pMD2.G) and Joel Watts (University of Toronto) for providing wild-type CAD5 and CAD5.Prnp^-/- cells.

## Author contributions

**Conceptualization:** Jean R. P. Gatdula, Robert C. C. Mercer, David A. Harris.

**Data curation:** Jean R. P. Gatdula, Jose Andres Alepuz Guillen.

**Formal analysis:** Jean R. P. Gatdula.

**Funding acquisition:** David A. Harris.

**Investigation:** Jean R. P. Gatdula.

**Methodology:** Jean R. P. Gatdula, Robert C. C. Mercer, Jose Andres Alepuz Guillen, Janelle S. Vultaggio.

**Project administration:** David A. Harris.

**Supervision:** David A. Harris.

**Validation:** Jean R.P . Gatdula.

**Visualization:** Jean R. P. Gatdula.

**Writing – original draft:** Jean R. P. Gatdula.

**Writing – review & editing:** Jean R. P. Gatdula, David A. Harris.

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
