## [Decision Letter · Decision Letter 0]

27 Aug 2025

Membrane-anchored PrPSc is the trigger for prion synaptotoxicity

PLOS Pathogens

Dear Dr. Harris,

Thank you for submitting your manuscript to PLOS Pathogens. After careful consideration, we feel that it has merit but does not fully meet PLOS Pathogens's publication criteria as it currently stands. Therefore, we invite you to submit a revised version of the manuscript that addresses the points raised during the review process.

Please submit your revised manuscript within 60 days Oct 26 2025 11:59PM. If you will need more time than this to complete your revisions, please reply to this message or contact the journal office at plospathogens@plos.org. Please include the following items when submitting your revised manuscript:

We look forward to receiving your revised manuscript.

Kind regards,

Julie A. Moreno, PhD

Guest Editor

PLOS Pathogens

Surachai Supattapone

Section Editor

PLOS Pathogens

Editor-in-Chief

PLOS Pathogens

orcid.org/0000-0003-2946-9497

Editor-in-Chief

PLOS Pathogens

orcid.org/0000-0002-7699-2064

**Journal Requirements:**

At this stage, the following Authors/Authors require contributions: Jean R.P. Gatdula, Robert C.C. Mercer, Janelle S. Vultaggio, and David A. Harris. Please ensure that the full contributions of each author are acknowledged in the "Add/Edit/Remove Authors" section of our submission form.

- ® on page: 20

- TM on pages: 17, 18, and 20.

5) We notice that your supplementary Figures are included in the manuscript file. Please remove them and upload them with the file type 'Supporting Information'. Please ensure that each Supporting Information file has a legend listed in the manuscript after the references list.

Potential Copyright Issues:

i) Figures Graphical Abstract, and S5A. Please confirm whether you drew the images / clip-art within the figure panels by hand. If you did not draw the images, please provide (a) a link to the source of the images or icons and their license / terms of use; or (b) written permission from the copyright holder to publish the images or icons under our CC BY 4.0 license. Alternatively, you may replace the images with open source alternatives. See these open source resources you may use to replace images / clip-art:

7) We note that your Data Availability Statement is currently as follows: "All data supporting the findings of this study are available within the article and its supplementary information files.". Please confirm at this time whether or not your submission contains all raw data required to replicate the results of your study. Authors must share the “minimal data set” for their submission. PLOS defines the minimal data set to consist of the data required to replicate all study findings reported in the article, as well as related metadata and methods (https://journals.plos.org/plosone/s/data-availability#loc-minimal-data-set-definition).

**Reviewers' Comments:**

Reviewer's Responses to Questions

**Part I - Summary**

Reviewer #1: In their manuscript, “Membrane-anchored PrPSc is the trigger for prion synaptotoxicity”, Gatdula and colleagues investigated the rapid onset of synaptotoxicity in primary hippocampal neurons following prion infection. The principal question the authors seek to address is whether synaptotoxicity is elicited by direct interaction between administered prions and PrPc-expressing neuronal cultures, or whether instead, prion protein conversion is a prerequisite for toxicity.

To address the nature of PrPc-PrPSc interaction that leads to dendritic spine retraction in primary cells, the authors investigate two distinct experimental paradigms aimed at assessing both the importance of interactions between prion seeds and surface-expressed PrPc, and the role of PrP conversion.

In their first approach, the authors examined whether expression of the protective prion protein variants G126V PrP and V208M PrP affects synaptotoxicity in primary neuronal cells challenged with distinct prion strains (RML, 22L and Me7). To address this, they transduced primary neuronal cells from Prnp-/- mice with the respective PrP variants and subsequently infected the cultures with prions (Figure 2). While expression of wild-type mouse PrP induced significant dendritic spine retraction (p < 0.0001), expression of G126V PrP did not elicit spine retraction in response to 22L and Me7 prions (p-value > 0.05), and showed only a modest effect for RML prions (p < 0.05) (Figure 2). In contrast, expression of V208M PrP variant did not prevent synaptotoxicity induced by any of the tested prion strain. In complementary prion replication assays using the N2a and Cad5 mouse lines, the authors assessed effects of protective prion protein variant expression on prion replication. They found that G126V PrP expression rendered cells refractory to all prion strains tested, whereas V208M PrP expression resulted in strain-dependent effects (Figure. 1). To corroborate that synaptotoxicity is associated with PrP conversion in primary neurons, the authors presented a statistically limited experiment showing that prion aggregates were absent in neurons expressing G126V PrP, but not in those expressing wild-type mouse PrP (Figure 3).

In their second experimental approach, the authors introduce a mouse/hamster species barrier to examine how seed-substrate incompatibility affects prion propagation and synaptotoxicity, respectively. Using their prion propagation model, they first show that 263K hamster prions are propagated exclusively in hamster PrPc expressing Cad5 cells, whereas 22L prions replicate only in mouse PrPc expressing cells (Figure 4). They then assess the impact of a species barrier in their spine retraction assay. Expression of hamster PrPc in mouse Prnp-/- neurons markedly reduced synaptotoxicity compared to expression of mouse PrPc. Conversely, expression of mouse PrPc confers substantial protection against synaptotoxic effects triggered by 263K hamster prions (Figure 5).

This is a well-designed and -controlled study that harnesses the groups advancement in establishing sensitive experimental assays for synaptotoxicity in primary cell models by monitoring changes in the branched actin filaments that precede spine retraction (PMID: 27227882, PMID: 30235355). This study takes advantage of two distinct cell models, primary hippocampal neurons and immortalized cell lines, N2a and Cad5, to examine and validate effects of protective prion protein variants (G126V and V208M PrP) and incompatibilities between substrate (PrP) and prion strain (263K versus 22L), respectively. By using the Cad5 prion propagation model, the authors confirm that expression of G126V PrP, but not that of V208M rendered cells refractory to all mouse-adapted prion strains tested (Figure 1). Using the same model, they go on to demonstrate that hamster PrP-expressing cells propagate hamster 263K prions, but not mouse 22L prions. In analogy, mouse PrP-expressing cells propagate 22L, but not 263K (Figure 4). Based on this validation, the authors then transduce primary hippocampal neurons from Prnp-/- mice with wild-type mouse, protective (G126V) or wild-type hamster PrP (Figures 3 and 5) and demonstrate that synaptotoxicity is associated with PrP conversion.

In supplementary figures, the authors provide further experimental controls to show

(i) that reconstitution of N2a Prnp-/- cells with wild-type PrP gives rise to cells and subclones that propagate 22L, RML, but not Me7 (Figure S2),

(ii) that dendritic spine retraction in hippocampal neurons is PrP-dependent and elicited by all prion strains tested, i.e. RML, 22L and Me7 (Figure S4),

(iii) that lentiviral transduction of hippocampal neurons has no effect on dendritic spine density (Figure S5),

(iv) that hamster 267K prions are not replicated in hamster PrP expressing N2a cells (Figure S7). That hamster 267K prions are replicated in hamster PrP expressing Cad5 cells is shown in Figure 4.

Reviewer #2: The mechanisms underlying prion-induced neurotoxicity remain poorly understood. The authors have made significant contributions in this area through a series of seminal studies, and this manuscript builds upon that findings by proposing that membrane attachment of PrPSc is a critical trigger for prion-induced synaptotoxicity. The manuscript is very well prepared, and the western blots and cell culture images are particularly convincing. The study addresses an important question and presents compelling data; however, several issues should be addressed to further strengthen the conclusions:

Reviewer #3: The manuscript titled “Membrane-anchored PrPSc is the trigger for prion synaptotoxicity” (PPATHOGENS-D-25-01703) by Gatdula et al. is a well-written and interesting study.

Prion infection leads to the death of infected neurons, but the molecular causes for this toxicity are still incompletely understood. Previous studies demonstrated the need to have both PrPC and PrPSc expressed on the same cells for the toxic effects to manifest. In the current study the authors express PrP molecules with known protective polymorphisms to investigate their effects on prion-related synaptotoxicity and how membrane-anchoring may play a role using a series of cell culture experiments.

**Part II – Major Issues: Key Experiments Required for Acceptance**

Reviewer #1: It remains unclear how Figure S6 relates to the overall finding of the study as expression of wild-type PrP in hippocampal neurons of Prnp-/- mice challenged with 22L prions does not yield visible puncta, in contrast to the images presented in Figure 3. If limited magnification accounts for the absence of puncta, this should be explicitly stated, and higher-magnification images should be provided. Figure 3 shows puncta in hippocampal neurons from Prnp-/- mice reconstituted with wild-type, but not in those expressing G126V PrP. While this finding is consistent with results obtained from the propagation model, the statistical power of the experiment is limited (“10 neurons from 2 independent experiments were visualized in randomly selected areas”). A larger number of independent replicated experiments is recommended to support the conclusion that PrP conversion is a prerequisite for synaptotoxicity.

Reviewer #2: To more precisely demonstrate that synaptotoxicity is dependent on membrane-anchored PrPSc, it would be valuable to include hippocampal neurons expressing GPI-anchorless wild-type PrP in the assay. This parallel experiment would help clarify whether membrane anchorage is indeed necessary for the observed toxic effects.

Reviewer #3: However, the membrane-anchoring effects are inferred only based on microscopical images, which are suggestive but not definitive. Controls expressing anchorless variants of wild-type PrP and its known protective polymorphisms would be needed to make this study truly convincing. While it may be possible to predict the outcome of such experiments, it would still be worthwhile to include such controls.

**Part III – Minor Issues: Editorial and Data Presentation Modifications**

Reviewer #1: While the background and the aim of the study is well described in the introduction, the abstract requires revision. In line 17-20, page 2, the authors summarize the background of the study: “However, these studies did not distinguish the source of the PrPSc that activates the signaling pathway: extracellular PrPSc bound to PrPc on the neuronal surface, or membrane-anchored PrPSc generated by the PrPc-PrPSc conversion process.” The phrase lacks clarity in articulating the primary aim of the study and the term “source of the PrPSc” is confusing. The primary aim of the study, as I understand it, is to determine whether synaptotoxicity is directly associated with PrP conversion or occurs independently of it. On page 5, line 87, the authors phrase the objective of the study in a clearer way: “To determine whether conversion of cell-surface PrPc to PrPSc is essential for prion neurotoxicity, we have employed two different strategies.”

Reviewer #2: 1. While N2a-KO and CAD5-KO cells demonstrate that the G126V and V208M mutations do not support prion replication, it remains unclear whether cultured hippocampal neurons share similar responses to these strains. To reinforce this point, the authors should provide evidence that these prion protein variants do not support prion replication in hippocampal neurons as well. If western blotting is limited by sample availability, immunohistochemistry (IHC) or other sensitive detection methods (RT-QuIC or PMCA) could be used to confirm the accumulation of PrPSc.

2.Regarding the second experimental strategy, the use of a cross-species (mouse-hamster) models with a strong species barrier, likely preventing prion replication due to incompatibility between the incoming prion seeds and the recipient PrP. Therefore, the absence of newly synthesized PrPSc under these conditions cannot be taken as evidence that synaptotoxicity requires membrane-bound prions.

Reviewer #3: A few minor items also need to be addressed:

1) In the introduction the authors write “… the removal of the GPI anchor of PrPC fundamentally alters prion pathogenesis, even though the unanchored PrPC can still function as a substrate to bind and propagate PrPSc. Taken together, these previous studies suggest that PrPC on the neuronal cell surface is essential for generating a neurotoxic signal in response to exogenous PrPSc.” However, a study by Stöhr et al. (2011) demonstrated the spontaneous emergence of PrPSc in mice overexpressing anchorless PrP, including neurotoxic effects. Therefore, these statements would need to be moderated.

2) On line 179 the jargon term “primary sequence” is used. It should be replaced by the proper term “primary structure”.

3) On lines 181-182 the species barrier between mouse and hamster PrP is attributed to the 12 residues difference in the primary structure. However, in the very next paragraph it is stated that murine N2a cells transfected with hamster PrP still cannot propagate hamster prions (263K) and that other factors seem to be at play. This apparent mismatch in statements needs to be straightened out.

4) The actin loading control in Figure 1D looks a bit uneven / not convincing.

5) In Figure 4A it would make it easier to understand the figure if the vertical labels “D18 (PrP)” and “3F4 (PrP)” would be swapped.

6) In Figure S2 the labels “N2Ap” (panel B) and “N2ap” (panel E) are not defined. Presumably they refer to N2A parent (mentioned in panel D).

7) Figure S5A shows glia cells which are not mentioned in the figure legend.

PLOS authors have the option to publish the peer review history of their article (what does this mean? ). If published, this will include your full peer review and any attached files.

**Do you want your identity to be public for this peer review?** For information about this choice, including consent withdrawal, please see our Privacy Policy .

Reviewer #1: No

Reviewer #2: No

Reviewer #3: No

**Figure resubmission:**

**Reproducibility:**



---

## [Decision Letter · Decision Letter 1]

19 Jan 2026

Dear %TITLE% Harris,

We are pleased to inform you that your manuscript 'Membrane-anchored PrPSc is the trigger for prion synaptotoxicity' has been provisionally accepted for publication in PLOS Pathogens.

Best regards,

Surachai Supattapone

Section Editor

PLOS Pathogens

Sumita Bhaduri-McIntosh

Editor-in-Chief

PLOS Pathogens

orcid.org/0000-0003-2946-9497

Michael Malim

Editor-in-Chief

PLOS Pathogens

orcid.org/0000-0002-7699-2064

Reviewer Comments (if any, and for reference):

Reviewer's Responses to Questions

**Part I - Summary**

Reviewer #1: The unmet need for addressing the mode of synaptotoxic signaling is now clearly articulated in the Abstract (lines 18f).

An increase in the number of independent experiments from two to four in Figure 3 is acceptable. In this experiment, the authors report a significantly higher number of puncta corresponding to newly converted PrPSc in 22L exposed, PrPC-reconstituted neurons. In contrast, no significant difference is observed in neurons reconstituted with G126V PrP. This underpins the suggestion of the authors that rapid PrP conversion at the plasma membrane is required for synaptotoxicity.

This reviewer has no further comments on the authors' responses to Reviewers 2 and 3.

Reviewer #2: Reviewed the revised manuscript, no futher comments. All my concerns have been addressed.

Reviewer #3: The revised manuscript titled “Membrane-anchored PrPSc is the trigger for prion synaptotoxicity” (PPATHOGENS-D-25-01703_R1) by Gatdula et al. is substantially improved over the initial submission.

**Part II – Major Issues: Key Experiments Required for Acceptance**

Reviewer #1: (No Response)

Reviewer #2: Reviewed the revised manuscript, no futher comments. All my concerns have been addressed.

Reviewer #3: none

**Part III – Minor Issues: Editorial and Data Presentation Modifications**

Reviewer #1: (No Response)

Reviewer #2: Reviewed the revised manuscript, no futher comments. All my concerns have been addressed.

Reviewer #3: none

PLOS authors have the option to publish the peer review history of their article (what does this mean? ). If published, this will include your full peer review and any attached files.

**Do you want your identity to be public for this peer review?** For information about this choice, including consent withdrawal, please see our Privacy Policy .

Reviewer #1: No

Reviewer #2: No

Reviewer #3: No

---

## [Editor Report · Acceptance letter]

Dear %TITLE% Harris,

We are delighted to inform you that your manuscript, "Membrane-anchored PrPSc is the trigger for prion synaptotoxicity," has been formally accepted for publication in PLOS Pathogens.

Best regards,

Sumita Bhaduri-McIntosh

Editor-in-Chief

PLOS Pathogens

orcid.org/0000-0003-2946-9497

Michael Malim

Editor-in-Chief

PLOS Pathogens

orcid.org/0000-0002-7699-2064